# Adolescents with Atopic Dermatitis Have Lower Peak Exercise Load Capacity and Exercise Volume Compared with Unaffected Peers

**DOI:** 10.3390/ijerph191610285

**Published:** 2022-08-18

**Authors:** Tsung-Hsun Yang, Po-Cheng Chen, Yun-Chung Lin, Yan-Yuh Lee, Yu-Hsuan Tseng, Wen-Hsin Chang, Ling-Sai Chang, Chia-Hsuan Lin, Ho-Chang Kuo

**Affiliations:** 1Department of Physical Medicine and Rehabilitation, Kaohsiung Chang Gung Memorial Hospital and Chang Gung University College of Medicine, Kaohsiung 833, Taiwan; 2Department of Sports Medicine, College of Medicine, Kaohsiung Medical University, Kaohsiung 807, Taiwan; 3Department of Pediatrics, Kaohsiung Chang Gung Memorial Hospital and Chang Gung University College of Medicine, Kaohsiung 833, Taiwan; 4Kawasaki Disease Center, Kaohsiung Chang Gung Memorial Hospital and Chang Gung University College of Medicine, Kaohsiung 833, Taiwan

**Keywords:** atopic dermatitis, adolescent, exercise, exercise test, motivation, self-efficacy

## Abstract

**Background:** Sweating and increased skin temperature caused by exercise can reduce physical activity and the willingness to exercise in adolescents with atopic dermatitis. This study was conducted to investigate the exercise load capacity of adolescents with atopic dermatitis and analyzed their exercise behavior and motivation. **Methods:** Adolescents with and without atopic dermatitis were assigned to the atopic dermatitis group and control group (*n* = 27 each). Both groups completed a cardiopulmonary exercise test and questionnaires to assess their exercise capacity, weekly exercise volume, exercise motivation, and self-efficacy, respectively. **Results:** The ratio of measured forced vital capacity to the predicted forced vital capacity and the peak oxygen consumption of the atopic dermatitis group were significantly lower than those of the control group. The Godin Leisure-Time Exercise Questionnaire scores of the atopic dermatitis group were significantly lower than those of the control group. As for the Behavioral Regulation in Exercise Questionnaire 2, the scores for the introjected and identified regulations of the atopic dermatitis group were significantly lower than those of the control group. Regarding the Multidimensional Self-Efficacy for Exercise Scale, the scheduling efficacy and total scores of the atopic dermatitis group were significantly lower than those of the control group. **Conclusions:** Adolescents with atopic dermatitis had lower peak exercise capacity and lower weekly exercise volume. Furthermore, they lacked the negative feelings toward inactivity and the self-confidence to plan regular exercise independently. The results of this study suggest that adolescents with atopic dermatitis should be encouraged to engage in regular indoor exercise.

## 1. Introduction

Atopic dermatitis (AD) is a chronic inflammatory skin disorder that usually occurs in early childhood. The pathogenic factors of AD, which results in immune dysregulation, ref. [1] are complex, with possible links to the interaction of genes, the immune system, and the environment. The main symptoms of AD include eczema of the face, neck, and limbs; pruritus; and recurrent inflammatory flare-ups [2].

Physical activity during adolescence can help to develop a healthy lifestyle that persists into adulthood and reduces the incidence of chronic illness [3]. However, exercise-induced sweating and increases in skin temperature can aggravate the symptoms of AD [4,5], reducing the willingness to exercise and the exercise volume in this patient population [6]. A lack of physical activity can negatively affect health in the long term, but few studies have surveyed the exercise behavior of adolescents with AD. Lim et al. reported that higher physical activity was positively associated with AD in Korean adolescents [7] but their assessment of exercise volume was limited. Kong et al. reported that within one year of diagnosis, Korean adolescents with AD did lower moderate and vigorous physical activity and muscle strengthening exercises compared to those without AD and that regular engagement with exercise had a positive impact on the mental health of this patient population [8]. No previous study has investigated the cardiopulmonary fitness of adolescents with AD. Additional investigation is required to understand the exercise behavior and exercise load capacity of adolescents with AD. The aim of this study was to investigate the exercise load capacity, regular exercise volume, motivation, and self-efficacy of adolescents with AD.

## 2. Materials and Methods

### 2.1. Experimental Design

This study adopted a cross-sectional design.

### 2.2. Participant Recruitment and Characteristics

Participant recruitment was conducted at a medical center in Taiwan. Patients diagnosed as having AD by a pediatric allergy specialist physician were assigned to the AD group (AD is defined as a chronic relapsing pruritic skin rash with dry skin, erythema, or scaling, as well as skin creases and characteristic areas [9,10]). Age- and gender-matched participants without any AD history or cardiopulmonary diseases that could affect function or physical activity were assigned to the control group. The inclusion criteria were patients with AD aged 13 to 19 years. Patients were excluded if they had (1) heart disease (e.g., heart failure, moderate-to-severe valvular heart disease, coronary artery disease, or severe arrhythmia), (2) lung disease (e.g., asthma or chronic obstructive pulmonary disease), (3) muscular, skeletal, or neurological diseases (e.g., fractures, arthritis, muscle strain, or polio) that could affect their ability to complete the exercise test, (4) obvious physical or mental disabilities and communication difficulties, or (5) refused to complete the exercise test or the questionnaires.

### 2.3. Exercise Habit and Environment

The participants were asked whether they engaged in regular outdoor or indoor exercise (at least 30 min per week). Participants in the AD group were asked whether exercise aggravated their AD symptoms and whether dermatitis affected their willingness to exercise.

### 2.4. AD Severity and Medication Status

The AD severity of the participants in the AD group was evaluated using the Scoring Atopic Dermatitis (SCORAD) Index [11], which consists of extent and intensity scores (total 83 points) and subjective symptom scores (total 20 points). AD symptom severity was determined using the extent plus intensity scores, where scores of <15, 15–40, and >40 corresponded to mild, moderate, and severe symptoms, respectively. The current dermatitis-related medication status (oral or topical steroids and the topical immunomodulator) of the AD group was also recorded.

### 2.5. Cardiopulmonary Exercise Test (CPET)

All participants were assessed using cardiopulmonary exercise testing (CPET) for the exercise load capacity. CPET was conducted using a MasterScreen CPX (Carefusion, 234 GmbH, Hochberg, Germany) to evaluate the participants’ cardiopulmonary function and exercise load capacity. Indicators of pulmonary function, including forced vital capacity (FVC), forced expiratory volume in 1 s (FEV1), and FEV1/FVC ratio, were evaluated through spirometry at rest. Subsequently, each participant was fitted with a mask connected to a gas analyzer and completed a treadmill exercise test according to the Bruce protocol [12]. The test was terminated if participants experienced unbearable symptoms [13]. During the test, blood pressure and heart rate were monitored and recorded. The gas analyzer recorded the oxygen uptake (VO_2_), exhaled carbon dioxide (VCO_2_), and minute ventilation (VE). When the respiratory exchange ratio (RER), the ratio of VCO_2_ to VO_2_, reached or exceeded 1.10, this demonstrated that sufficient exercise intensity had been achieved. The VE/VO_2_ and VE/VCO_2_ ratios can be employed to estimate the anaerobic threshold (AT). The ratio of oxygen consumption per kilogram of body weight (VO_2_/kg) at the AT and the predicted VO_2_/kg at the peak (hereafter referred to as AT%) were used to determine aerobic metabolic capacity. The ratio of VO_2_/kg at the peak and the predicted VO_2_/kg at the peak (hereafter referred to as Peak%) were used to assess peak exercise load capacity. Clinically, a Peak% of 85% or higher is considered a “normal” test result [14]. The peak rate pressure product (PRPP) can reflect myocardial perfusion and the stroke volume can be estimated from the ratio of the oxygen consumption to heart rate (O_2_ pulse). After the gender, age, height, and weight of the participants were entered, the predicted values of FVC, FEV1, VO_2_/kg at the peak, and O_2_ pulse at the peak were calculated and compared with the measured data.

### 2.6. Questionnaires

For assessment, participants completed self-reporting questionnaires including the Godin Leisure-Time Exercise Questionnaire (GLTEQ), the Chinese version of the Behavioral Regulation in Exercise Questionnaire 2 (BREQ-2), and the Multidimensional Self-Efficacy for Exercise Scale (MSES). The GLTEQ solicits responses concerning the average number of times high-, medium-, and low-intensity exercises were performed for more than 15 min per week in the previous year; the results thus reflect participants’ weekly exercise volumes. Higher scores represent a higher weekly exercise volume [15,16]. The BREQ-2 evaluates exercise motivation and behavioral regulation. It contains 5 aspects related to exercise behavior regulation, namely amotivation, external regulation, introjected regulation, identified regulation, and intrinsic regulation. Higher scores represent higher agreement with the item description [17,18]. MSES, which comprises 9 items, considers 3 types of self-efficacy related to exercise: task efficacy, coping efficacy, and scheduling efficacy [19]. On a scale from 0 to 10, respondents rate their level of confidence in the situation described, with higher scores indicating higher confidence [19].

### 2.7. Statistical Analysis

Analyses were performed using Medcalc software version 19 (Medcalc Software, Ostend, Belgium). The chi-square test was conducted to compare intergroup differences in the categorical variables (e.g., gender, regular exercise habit, reasons for CPET termination, and the achievement of sufficient exercise intensity). The independent samples *t*-test was performed to examine the continuous variables (e.g., age, height, weight, BMI, and CPET results). A *p* value of <0.05 was considered statistically significant.

## 3. Results

### 3.1. Demographics

The study period was from February 2019 to July 2020. In all, 54 participants (*n* = 27 in each group) were enrolled. No statistically significant differences in gender, age, height, weight, or body mass index between the two groups were observed (Table 1).

### 3.2. Severity of AD and Medication Status

The mean score of the SCORAD index in the AD group was 19.11 ± 13.91 points, and 12 (44%), 13 (48%), and 2 (7%) of the participants were classified as having mild, moderate, and severe AD, respectively (Table 1). Eight participants (30%) in the AD group did not regularly take medication for AD. Nineteen (70%) used at least one oral or topical medication regularly.

### 3.3. Exercise Environment and Willingness

In all, 25 participants (93%) in the AD group and 22 participants (81%) in the control group regularly engaged in outdoor exercise, and 12 (44%) in the AD group and 17 (63%) in the control group regularly engaged in indoor exercise. No significant difference was observed between the two groups in regular engagement in outdoor and indoor exercise (*p* = 0.229 and *p* = 0.176, respectively).

Overall, 15 of the 25 participants in the AD group who regularly engaged in outdoor exercise reported that outdoor exercise exacerbated their symptoms. Only 1 of the 12 participants who regularly engaged in indoor exercise reported that it exacerbated their symptoms. In total, 7 of the 27 participants noted that the symptoms of AD affected their willingness to exercise.

### 3.4. CPET Results

All participants completed the CPET. No adverse events occurred. In the AD group, 16, 6, 3, and 2 participants terminated their CPET because of soreness in their legs (59%), shortness of breath (22%), VO_2_ plateauing (11%), and perceived inability to match the speed of the treadmill (7%), respectively. Moreover, 13, 8, and 6 participants in the control group ended their tests because of soreness in their legs (48%), shortness of breath (30%), and VO_2_ plateauing (22%), respectively. A comparison of the reasons for test termination between the two groups revealed no significant difference (*p* = 0.309).

Table 2 presents a comparison of the results of lung function and CPET between the AD and control groups. FVC%, VO_2_/kg at peak, and Peak% were significantly lower in the AD group than in the control group. No significant between-group differences in other measures were identified.

### 3.5. Questionnaire Results

Table 3 presents an intergroup comparison of the questionnaire results. The GLTEQ scores, introjected and identified regulation scores on the BREQ-2, and scheduling efficacy and total MSES scores of the AD group were significantly lower than those of the control group. No other significant intergroup differences were identified.

## 4. Discussion

This is the first study to investigate the capacity, behavior, and motivation of exercise in adolescents with AD. Compared with the control group, the AD group had a lower VO_2_/kg at peak and Peak%, indicative of a lower peak exercise load capacity. According to the GLTEQ scores, the AD group had a lower weekly exercise volume. The results support the premise that adolescents with AD exercise less than their unaffected peers. Moreover, the results also provide evidence to show that this inactivity negatively affects their cardiopulmonary exercise capacity.

Various factors can lead to the lower physical activity of individuals with AD. As mentioned, sweating and high skin temperatures from exercise can aggravate symptoms [4]. Moreover, dermatitis on the soles or the palms can limit the types of sports that can be played [20]. However, regular exercise can improve physical fitness performance and bone growth, reduce psychological stress, and reduce the risk of cardiovascular disease [21,22,23]. A cross-sectional study by Kong et al. noted that patients with chronic AD who exercised regularly tended to have lower psychological stress [8]. In this study, considerably fewer participants reported the aggravation of AD symptoms from indoor exercise than from outdoor exercise (1/12 vs. 15/25; 8% vs. 60%). Thus, the researchers assumed that indoor exercise without sun exposure induced fewer exercise-related symptoms of AD. Based on these results, adolescents with AD should be encouraged to engage in indoor exercise for better health and lesser aggravation of AD symptoms [24].

In our study, FVC% was lower in the AD group than in the control group. FVC is the maximum volume of air exhaled with a maximally forced effort and could be reduced due to suboptimal patient effort, respiratory muscle weakness, restriction, airflow limitation, or a combination of these [25]. It could be a key indicator of long-term survival. Burney and Hooper revealed that FVC, and not FEV1 or FEV1/FVC, predicts survival in adults without chronic respiratory diagnosis or persistent respiratory symptoms [26]. However, to the best of our knowledge, no previous study has investigated FVC in patients with AD. Fuster et al. indicated that FVC increases with increasing physical activity [27]. In the present study, the lower FVC% of the AD group could have resulted from their low level of physical activity. Nonetheless, the status of FVC in patients with AD must be explored further in future studies.

The introjected and identified regulation scores of the AD group were lower than those of the control group, indicating that the participants with AD were less likely to have negative feelings regarding their lack of exercise. They also indicated that these participants failed to ascribe health benefits to regular exercise. The scheduling efficacy score and total score of the AD group on the MSES were also lower than those of the control group, demonstrating that the AD group had less confidence in effectively formulating an appropriate exercise plan. In the study by Kong et al., adolescents with chronic AD were less likely to engage in physical activity of moderate-to-high intensity [8], a trend that persisted with age. Silverberg and Greenland reported that the intensity, frequency, and amount of exercise in adults with a history of AD were lower than those in adults without AD [28]. Therefore, the cultivation of the positive value of exercise and regular exercise habits should begin in adolescence. In health education, the benefits of regular exercise should be explained to enhance the willingness to engage in exercise as well as the amount of exercise. Mobile devices and relevant software can be applied to monitor exercise volume as well as to set a reminder for regular exercise.

Multiple studies have supported the possible relationship between AD and exercise-induced bronchoconstriction (EIB) [29,30], and the impact of EIB could reduce participation in physical activity [31]. Diagnostic testing methods for EIB include measuring *lung function* changes after a standardized exercise challenge test [32]. However, because of equipment and technical constraints, we performed a pulmonary function test only once before exercise but not after exercise. Although the percentage of participants who ended their tests because of shortness of breath was lower in the AD group than in the control group (22% vs. 30%), EIB could not be excluded. Future research must be conducted to investigate the relationship between EIB, exercise load capacity, and physical activity.

This study has several limitations. First, the sample size was relatively small. Second, all the participants were patients at the same medical center and most of them had AD of mild-to-moderate severity; in other words, they may not be representative of all adolescents with this condition. Third, exercise behavior was assessed through the self-reporting GLTEQ. Self-reporting questionnaires are susceptible to problems concerning memory recall; moreover, they cannot reflect all physical activities performed in daily life. Fourth, repeat pulmonary function tests were not performed after exercise so EIB could not be excluded. Finally, Taiwan has a subtropical climate (except for the southern part of the island, which has a tropical climate) characterized by warmth and humidity, conditions that could affect the symptom severity of individuals with AD. Therefore, future studies should increase the sample size, include patients with varying severities of AD, and recruit individuals from different regions or countries.

## 5. Conclusions

In comparison with their unaffected peers, adolescents with AD had a lower peak exercise load capacity, FVC%, and weekly exercise volume, as well as fewer negative feelings toward inactivity, less positive recognition of the benefits of regular exercise, lower exercise self-efficacy, and less self-confidence in planning regular exercise. The results indicate that physicians and parents should encourage adolescents with AD to exercise indoors and to support their motivation and self-efficacy.

## Figures and Tables

**Table 1 ijerph-19-10285-t001:** General characteristics of the participants in each group.

	AD Group (*n* = 27)	C Group (*n* = 27)	*p*
Sex (male/female)	15/12	14/13	0.787
Age (year)	14.59 ± 1.85	15.37 ± 1.98	0.142
Height (cm)	163.59 ± 9.70	165.26 ± 9.80	0.533
Weight (kg)	63.11 ± 19.59	58.96 ± 13.45	0.369
BMI (kg/m^2^)	23.10 ± 4.88	21.53 ± 4.12	0.207
SCORAD index (score)	26.44 ± 16.21		
SCORAD index: extent + intensity (score)	19.11 ± 13.91		

Values are expressed as mean ± standard deviation. The chi-square test and independent samples *t*-test were used to compare differences between the two groups. AD: Atopic dermatitis; BMI: Body mass index; SCORAD: Scoring of atopic dermatitis.

**Table 2 ijerph-19-10285-t002:** Results of CPET in each group.

	AD Group (*n* = 27)	C Group (*n* = 27)	*p*
FVC (L)	2.99 ± 0.81	3.34 ± 0.79	0.117
FVC% (%)	80.24 ± 10.36	87.52 ± 11.37	0.017 *
FEV1 (L)	2.79 ± 0.78	3.02 ± 0.72	0.268
FEV1% (%)	89.00 ± 12.82	94.27 ± 14.40	0.162
FEV1/FVC (%)	93.19 ± 7.60	90.61 ± 9.16	0.265
VO_2_/kg at AT (mL/min/kg)	25.50 ± 7.34	27.68 ± 7.95	0.305
AT% (%)	61.35 ± 18.67	68.36 ± 17.95	0.169
VO_2_/kg at peak (mL/min/kg)	32.03 ± 8.77	37.15 ± 9.21	0.041 *
Peak% (%)	78.67 ± 20.73	90.81 ± 16.79	0.022 *
Peak% exceeded 85% (Yes/No)	11/16	17/10	0.106
Peak O_2_ pulse (mL/beat)	10.94 ± 3.65	11.89 ± 3.27	0.320
O_2_ pulse% (%)	91.48 ± 19.31	97.48 ± 21.08	0.281
RER at peak	1.16 ± 0.11	1.21 ± 0.12	0.142
RER < 1.10 (Yes/No)	7/20	3/24	0.165
PRPP	30,289.59 ± 4187.69	31,288.22 ± 4465.42	0.401

Values are expressed as mean ± standard deviation. The independent samples *t*-test was used to compare differences between the two groups. CPET: Cardiopulmonary exercise test; AD: Atopic dermatitis; FVC: Forced vital capacity; FVC%: Percentage of FVC compared with predicted FVC; FEV1: Forced expiratory volume in one second; FEV1%: Percentage of FEV1 compared with predicted FEV1; VO_2_: Oxygen uptake; AT: Anaerobic threshold; AT%: Percentage of VO_2_/kg at AT compared with predicted peak VO_2_/kg; Peak%: Percentage of VO_2_/kg at peak compared with predicted peak VO_2_/kg; O_2_ pulse%: Percentage of peak O_2_ pulse compared with predicted peak O_2_ pulse; RER: Respiratory exchange ratio; PRPP: Peak Rate-Pressure Product. * *p* < 0.05.

**Table 3 ijerph-19-10285-t003:** Results of Godin Leisure-Time Exercise Questionnaire, BREQ-2, and Multidimensional Self-Efficacy for Exercise Scale in each group.

	AD Group (*n* = 27)	C Group (*n* = 27)	*p*
Godin Leisure-Time Exercise Questionnaire (score)	23.85 ± 23.27	43.04 ± 33.19	0.017 *
BREQ-2			
Amotivation (score)	3.26 ± 3.43	3.22 ± 3.84	0.970
External regulation (score)	4.52 ± 3.99	4.00 ± 3.32	0.606
Introjected regulation (score)	2.07 ± 2.84	4.26 ± 3.18	0.010 *
Identified regulation (score)	5.81 ± 3.66	8.89 ± 3.31	0.002 *
Intrinsic regulation (score)	10.85 ± 3.69	11.89 ± 4.13	0.335
RAI (score)	23.30 ± 28.74	31.52 ± 28.12	0.281
Multidimensional Self-Efficacy for Exercise Scale			
Task efficacy (score)	20.56 ± 6.64	20.67 ± 5.35	0.946
Coping efficacy (score)	9.67 ± 6.15	10.85 ± 5.91	0.474
Scheduling efficacy (score)	12.11 ± 9.16	19.74 ± 8.07	0.002 *
Total score (score)	42.33 ± 16.28	51.89 ± 16.28	0.036 *

Values are expressed as mean ± standard deviation. The independent samples *t*-test was used to compare differences between the two groups. BREQ-2: The Behavioral Regulation in Exercise Questionnaire, 2nd edition; AD: Atopic dermatitis; RAI: Relative autonomy index. * *p* < 0.05.

## Data Availability

The data presented in this study are available on request from the corresponding author. The data are not publicly available due to privacy.

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
