# Peer review of "Adolescents with Atopic Dermatitis Have Lower Peak Exercise Load Capacity and Exercise Volume Compared with Unaffected Peers"

_ijerph, 2022, doi:10.3390/ijerph191610285_

Round 1

Reviewer 1 Report

The weakness of this study is the few number of cases (n = 27), as stated in the discussion. A small sample size may make the result questionable. It seems that AD subjects are shorter with more weights, despite the statistical insignificance for comparisons between both groups. The features may affect the pulmonary function for AD subjects in the first place. Some covariance analysis should be conducted to adjust for covariance like age, height, weight, etc.

The result of differences in pulmonary function should be discussed a bit further. Are the lower results for AD group normal in reference to standards? 

Reviewer 2 Report

Lack of physical activity is reduced by symptoms of atopic dermatitis, and health may be negatively affected in adolescents with atopic dermatitis. Then exercise load capacity, regular exercise volume, motivation, and self-efficacy in adolescents with atopic dermatitis were investigated because of no previous study. From the results of cardiopulmonary exercise testing, values of FVC% and VO2/kg at peak weakly in adolescents with atopic dermatitis were significant from those in control. From the results of questionnaire, scores of introjected regulation, identified regulation, scheduling efficacy and total scores were also significant in adolescents with atopic dermatitis were also significant. Then significant results of lower peak exercise load capacity, FVC%, and weakly exercise volume in adolescents with atopic dermatitis as well as fewer negative feeling toward inactivity, less positive recognition of the benefits of regular exercise, lower exercise self-efficacy, and less self-confidence in planning support encouragement of physicians and parents. 

Author Response

Thank you for your comments.

Round 2

Reviewer 1 Report

Questions are addressed accordingly.